# The Role of Tau in Neuronal Function and Neurodegeneration

**DOI:** 10.3390/neurolint17050075

**Published:** 2025-05-13

**Authors:** Gonzalo Emiliano Aranda-Abreu, Fausto Rojas-Durán, María Elena Hernández-Aguilar, Deissy Herrera-Covarrubias, Luis Isauro García-Hernández, María Rebeca Toledo-Cárdenas, Donají Chi-Castañeda

**Affiliations:** Instituto de Investigaciones Cerebrales, Universidad Veracruzana, Av. Luis Castelazo Ayala s/n, Col. Industrial Animas, Xalapa 91190, Mexico; frojas@uv.mx (F.R.-D.); elenahernandez@uv.mx (M.E.H.-A.); dherrera@uv.mx (D.H.-C.); rtoledo@uv.mx (M.R.T.-C.); lchi@uv.mx (D.C.-C.)

**Keywords:** Tau protein, tauopathies, Alzheimer’s disease, neurodegeneration, microtubules, glial cells

## Abstract

Tau protein plays a pivotal role in maintaining neuronal structure and function through its regulation of microtubule stability and neuronal polarity. Encoded by the *MAPT* gene, Tau exists in multiple isoforms due to alternative mRNA splicing, with differential expression in the central and peripheral nervous systems. In healthy neurons, *tau* mRNA is selectively localized and translated in axons, a process tightly regulated by untranslated regions (UTRs) and RNA-binding proteins such as HuD and FMRP. Pathologically, Tau undergoes hyperphosphorylation, misfolding, and aggregation, which contribute to neurodegeneration in a range of disorders collectively known as tauopathies. Alzheimer’s disease (AD) is the most prevalent tauopathy, where abnormal Tau accumulation in the temporal and frontal lobes correlates with cognitive decline and behavioral symptoms. Other tauopathies, including Progressive Supranuclear Palsy (PSP), Corticobasal Degeneration (CBD), Frontotemporal Dementia with Parkinsonism (FTDP-17), and Pick’s disease, are distinguished by the predominance of specific Tau isoforms (3R or 4R), cellular distribution, and affected brain regions. Notably, astroglial tauopathies highlight the pathological role of Tau accumulation in glial cells, expanding the understanding of neurodegeneration beyond neurons. Despite advances in imaging biomarkers (e.g., Tau-PET) and molecular diagnostics, effective disease-modifying therapies for tauopathies remain elusive. Ongoing research targets Tau through immunotherapies, splicing modulators, kinase inhibitors, and antisense oligonucleotides, aiming to mitigate Tau pathology and its deleterious effects. Understanding the multifaceted roles of Tau in neuronal and glial contexts is critical for developing future therapeutic strategies against tauopathies.

## 1. *Tau* Expression

Neurons in vertebrates exhibit a highly defined structural polarity, characterized by the presence of multiple branched dendrites and a single, typically unbranched, long axon. This organization reflects the functional specialization of their components, where dendrites play a primary role in signal reception, while the axon is responsible for transmitting these signals to other neurons. The extensive branching of dendrites maximizes the surface area available for forming synapses and receiving information from presynaptic neurons. In contrast, the axon, being an elongated and sparsely branched structure, ensures the efficient and directed conduction of action potentials toward postsynaptic neurons [1]. This structural arrangement is fundamental to neuronal physiology and polarity, enabling precise and effective communication within the neural circuits of the nervous system.

Microtubules play an essential role in the formation and maintenance of neuronal polarity, a critical process for neuronal function. This role has been demonstrated through ultrastructural studies that have revealed their organization in both axons and dendrites [2]. In axons, microtubules are oriented unidirectionally with their plus ends facing the terminal region, facilitating the efficient transport of components toward the axon terminals. In contrast, dendritic microtubules display a mixed organization, with both plus- and minus-end orientations, contributing to the unique dynamics and functionality of these structures [3].

The stabilization of microtubules is mediated by specific proteins that vary depending on the neuronal region. In dendrites, the microtubule-associated protein MAP2 is responsible for maintaining their stability, while in axons, this function is performed by the Tau protein [4]. These proteins not only stabilize microtubules but also support the functional specialization of the different regions of the neuron, ensuring effective intracellular transport and the preservation of neuronal polarity.

The Tau protein family belongs to the group of microtubule-associated proteins (MAPs) and is characterized by its heterogeneity in size and function. These proteins are encoded by the *MAPT* gene, located on human chromosome 17q21.31, and their diversity arises from the alternative splicing process of mRNA, giving rise to several isoforms with different molecular weights [5].

In the human central nervous system, there are six main isoforms of *tau*, which differ in the presence or absence of certain exons (exons 2, 3, and 10). These isoforms differ in the number of microtubule-binding repeats—three (3R) or four (4R)—which also influences their total amino acid length and, consequently, their molecular weight [6].

*Tau* 3R is an isoform that lacks exon 10, and its molecular weight ranges between 37 and 39 kDa, depending on the inclusion of exons 2 and 3. *Tau* 4R is an isoform that includes exon 10, and its molecular weight ranges between 65 and 70 kDa [5].

In the peripheral nervous system, additional *tau* isoforms exist that contain exon 4A, with a molecular weight of 110 kDa. These isoforms are known as “Big Tau” [7]. Table 1.

The human *tau* gene promoter contains multiple transcription factor binding sites, including elements that respond to neuronal signals, such as SP1 (a tissue-specific transcription factor) and others associated with the regulation of neuronal development and differentiation [8,9]. It lacks a classical TATA box region, typical of constitutively expressed genes, but contains a GC-rich region common in active promoters [10]. Its regulatory elements include NRF2, a transcription factor involved in the antioxidant response. The promoter contains an NRF2 binding site in the first intron, which may influence the inclusion of exon 3, associated with a potential protective role under conditions of cellular stress [11]. CREB, a cAMP response element-binding protein, has also been identified among the regulators of *MAPT* gene expression, suggesting that Tau expression may be modulated by cAMP-dependent signaling pathways, especially in processes of neuronal plasticity [12].

## 2. Localization of Tau mRNA

*Tau* messenger RNA (mRNA) is an essential component in regulating the synthesis of the Tau protein at specific locations within the neuronal cell, particularly in the axon. The precise localization of *tau* mRNA allows for the in situ synthesis of the protein, a crucial process for maintaining microtubule stability and neuronal polarity. This process is mediated by specific elements located in the untranslated regions (UTRs) of the mRNA and by associated proteins that facilitate its transport, localization, and translation [13,14].

*Tau* mRNA is transcribed in the nucleus and, after processing, is exported to the cytoplasm, where it associates with regulatory proteins to form ribonucleoprotein complexes (RNPs). These RNPs are transported along microtubules toward the axons by motor proteins, such as kinesins, which move toward the plus ends of microtubules. Once in the axon, Tau mRNA is anchored at specific sites near the microtubules to ensure localized translation. This process allows Tau protein to be synthesized directly where it is needed to stabilize the cytoskeleton [15,16].

*Tau* mRNA is associated with proteins such as RNA-binding proteins (RBPs). HnRNP A2/B1 recognizes specific sequences in the 3′ UTR of *tau* mRNA and facilitates its transport to the axon [17]. FMRP (Fragile X Mental Retardation Protein) associates with *tau* mRNA in RNPs, regulating both its transport and localized translation [18]. Staufen helps stabilize *tau* mRNA at specific locations within the axon [19,20].

Motor proteins such as Kinesins and Dyneins are involved in the directional transport of *tau* mRNA along microtubules [21]. Translational regulators such as proteins like PABP (Poly(A)-binding protein) and initiation factors of translation regulate the activation of protein synthesis in specific locations [22].

The untranslated regions (UTRs) of *tau* mRNA, at the 5′ and 3′ ends, contain cis-regulatory elements, with a uracil-rich region of 240 bases [14] that is essential for its transport, localization, and translation regulation. The 5′ UTR contains elements that regulate the efficiency of mRNA translation. It is a key site for interaction with translation initiation factors that control the onset of protein synthesis in the axon. The 3′ UTR is particularly important for the localization of *tau* mRNA [23], and includes Axonal Localization Sequences (ALS) that serve as signals for proteins such as HnRNP A2/B1 to recognize and transport mRNA to the axon [24]. Motifs for microRNA binding—such as those recognized by miR-132—modulate the stability and translational efficiency of tau mRNA in response to neuronal signaling cues [25]. Response elements to these stimuli located in the 3′ UTR allow *tau* mRNA localization and translation to be modulated by extracellular factors, such as BDNF or NGF, through intracellular signaling cascades [14,26]. Polyadenylation of the poly(A) tail in the 3′ UTR influences the stability and translational capacity of the mRNA. Proteins such as PABP bind to this tail to regulate its interaction with the translation machinery [27].

Another important protein is HuD (also known as ELAVL4, Embryonic Lethal Abnormal Vision-Like 4), an RNA-binding protein (RBP) that belongs to the Hu/ELAV family [28,29]. This protein is involved in the post-transcriptional regulation of gene expression in neurons, where it plays a key role in the stabilization, transport, and translation of specific mRNAs, including those encoding proteins related to neuronal growth, plasticity, and survival. HuD contains three RNA Recognition Motif (RRM)-type RNA-binding domains, which allow it to specifically interact with adenine- and uracil-rich sequences (AREs, AU-rich elements) present in the 3′ UTR regions of certain mRNAs, including *tau* [30].

These domains facilitate its interaction with mRNA and with other components of the transport and translation machinery. Although Hu/ELAV proteins are present in different tissues, HuD is primarily expressed in the nervous system, where it is associated with processes such as neuronal differentiation and regeneration. HuD is found both in the nucleus and cytoplasm, depending on the cellular context and stage of neuronal development. In the cytoplasm, it associates with mRNA in transport or active translation [31].

The main functions of HuD include mRNA stabilization, where it binds to ARE regions in the 3′ UTR of specific mRNAs, increasing their stability by protecting them from ribonuclease-mediated degradation [32]. This is particularly important in neurons, where mRNA stability regulates the prolonged expression of essential proteins. In mRNA transport, HuD participates in the formation of ribonucleoprotein complexes (RNPs) that transport mRNAs to specific regions of neurons, such as dendrites and axons [16]. This transport is essential for localized protein synthesis required for synaptic plasticity and axonal regeneration. HuD regulates mRNA translation by facilitating access to the translation machinery or by inhibiting it in temporary repositories. This mechanism ensures that protein synthesis occurs in a controlled manner and in response to neuronal stimuli. HuD binds to AU-rich elements present in the 3′ UTR of *tau* mRNA. This interaction facilitates the stabilization of *tau* mRNA, prolonging its half-life and allowing its accumulation in regions where Tau is needed [33]. In the axon, HuD collaborates in the transport of Tau mRNA toward the axon, contributing to the localized synthesis of Tau protein for microtubule stabilization. HuD participates in the regulation of *tau* mRNA translation in response to local stimuli, ensuring that Tau synthesis occurs at the appropriate time and place [28,34].

## 3. Tau and Neurodegenerative Diseases

Disorders such as progressive supranuclear palsy (PSP), corticobasal degeneration (CBD), and related conditions are collectively classified as tauopathies due to the pathological accumulation of Tau protein.

### 3.1. Alzheimer’s Disease

Alzheimer’s disease (AD) is the most common form of dementia and a major cause of disability in older adults. It is clinically characterized by progressive memory loss, cognitive decline, and behavioral changes. Affecting approximately 50 million people globally, its prevalence increases significantly with age, reaching 10–20% among individuals over 65. AD is primarily associated with two hallmark pathological features: extracellular accumulation of β-amyloid (Aβ) plaques and intracellular neurofibrillary tangles formed by hyperphosphorylated Tau protein. While Aβ plaques are deposited in the brain’s extracellular space, Tau tangles accumulate within neurons, contributing significantly to neuronal degeneration, particularly in the temporal and frontal lobes [35].

Tau pathology begins in the entorhinal cortex and spreads to the hippocampus and temporal lobe, following the pattern described in Braak staging [36,37]. Tau-PET imaging with tracers such as flortaucipir reveals a strong spatial correlation between tracer uptake and Tau pathology in the hippocampus, entorhinal cortex, and medial/superior temporal gyri [38].

The temporal lobe, essential for memory, language, and sensory processing, is a key region affected in AD. Elevated Tau levels in the left temporal lobe have been associated with language deficits, while right-lobe accumulation correlates with behavioral and emotional disturbances [39,40,41]. This asymmetric distribution of Tau, confirmed in PET studies of 695 individuals, may serve as a diagnostic marker [41].

Neuroimaging and histopathological evidence show that pTau levels correlate more strongly with cortical thinning in the medial temporal lobe (MTL) than Aβ deposition, suggesting that cortical thickness is a reliable in vivo biomarker of Tau pathology [42,43,44]. Moreover, Tau accumulation in these regions corresponds with clinical severity and is confirmed by autopsy studies [45].

Neuroinflammatory processes contribute to Tau-related degeneration. Increased levels of soluble Triggering Receptor Expressed on Myeloid cells 2 (sTREM2), a marker of microglial activation, have been associated with Tau burden and neurodegeneration independent of Aβ levels. This mechanism also overlaps with Primary Age-Related Tauopathy (PART), which affects the medial temporal lobe and is exacerbated by microglial activity [46].

In addition, astrocytic responses are evident in AD. Elevated levels of GFAP, AQP-4, GLT-1, and Kir4.1 have been detected in the middle temporal gyrus. Redistribution of these proteins, such as the extension of Kir4.1 into astrocytic processes, reflects glial involvement in Tau-mediated damage [47].

Tau pathology also affects the frontal lobe, a region responsible for executive functions, emotion, and behavior. In AD, hyperphosphorylated Tau (pS396) in the frontal cortex is associated with neuropsychiatric symptoms such as agitation and aggression, evaluated through the Present Behavioral Examination (PBE). These changes coincide with increased GSK-3β expression and reduced PP2A activity [48].

Interestingly, while Tau levels may decrease in late AD stages due to neuronal loss, pTau continues to rise [49]. Structural and metabolic alterations also occur early in the disease: executive dysfunction measured by the Frontal Assessment Battery (FAB) correlates with reduced gray matter volume and glucose metabolism in the right inferior frontal gyrus [50].

Finally, high molecular weight Tau oligomers (HMWoTau) have been identified across all Braak stages. These oligomers appear early (stages I–III) and increase dramatically in advanced stages (V–VI), highlighting their role in neurodegeneration [51].

### 3.2. Progressive Supranuclear Palsy (PSP)

Progressive Supranuclear Palsy (PSP) is a rare neurodegenerative disease that affects mobility, cognition, and eye function [52]. PSP is estimated to affect between 5 and 7 people per 100,000 inhabitants, with increasing prevalence with age, typically appearing between the ages of 60 and 70 [53].

Among its clinical manifestations, PSP is distinguished by vertical gaze palsy, with difficulties moving the eyes up or down in most cases. Additionally, patients often experience early and unexplained falls, which is a hallmark of the progression of the disease. Bradykinesia and rigidity are other common symptoms; however, unlike Parkinson’s disease, rigidity in PSP is predominantly axial and poorly responsive to levodopa. Moreover, over 50% of patients exhibit cognitive impairment, affecting decision-making, planning, and impulse control. In advanced stages, dysphagia and dysarthria significantly reduce quality of life, leading to complications such as aspiration pneumonia [54].

Tau protein is essential for microtubule stability in neurons. In PSP, Tau undergoes pathological modifications such as hyperphosphorylation, promoting its aggregation and subsequent neuronal degeneration [55]. Specifically, Tau aggregates in PSP are predominantly composed of the 4R isoform, differentiating it from other tauopathies. These abnormal accumulations particularly affect the substantia nigra, thalamus, midbrain, and frontal cortex [56].

The diagnosis of PSP is clinical and supported by auxiliary tools such as the criteria from the Movement Disorder Society (MDS-PSP, 2017), brain MRI, and positron emission tomography (PET), which can identify specific patterns of atrophy and hypometabolism [57]. The presence of the “hummingbird sign” on MRI, characterized by midbrain atrophy with preserved pons, is a distinctive finding. Additionally, cerebrospinal fluid biomarkers, such as elevated levels of phosphorylated Tau and decreased β-amyloid 42, have proven useful in differentiating PSP from Alzheimer’s disease [58].

Currently, there is no cure for PSP, and available treatments aim to improve patients’ quality of life. Symptomatic approaches include the use of levodopa and dopaminergic agonists, though with limited efficacy. Other drugs such as amantadine may help reduce rigidity and apathy. Physical therapy, occupational therapy, and speech therapy are essential to delay functional disability and prevent complications [59].

In the research field, various therapeutic strategies are under development. Anti-Tau immunotherapy, based on monoclonal antibodies such as Gosuranemab and Tilavonemab, has shown a reduction in Tau levels in the CSF, although without significant clinical impact so far [60]. Other strategies include Tau aggregation inhibitors like LMTX (modified methylene blue) and Tau splicing modulators to reduce the production of the 4R-Tau isoform. In addition, inhibition of the GSK3β kinase, involved in Tau phosphorylation, represents a promising line of research [61].

The prognosis of PSP is guarded, with an average survival of 5 to 9 years from the onset of the first symptoms. Disease progression leads to increasing dependence on daily activities, significantly impacting the quality of life of both patients and caregivers. Research on PSP continues to advance, aiming to improve early detection and develop disease-modifying treatments. Understanding the biology of Tau and exploring neuroinflammation as a therapeutic target are key areas for the future of research in this condition.

### 3.3. Tau and Its Role in CBD

Corticobasal Degeneration (CBD) is a rare neurodegenerative disease that affects movement, cognition, and sensory function [62]. The prevalence of CBD is low, affecting approximately 4–5 people per 100,000 inhabitants, and it typically appears between the ages of 60 and 70 [62].

From a clinical standpoint, CBD is distinguished by a combination of motor, cognitive, and sensory symptoms that progress asymmetrically. Among its most characteristic manifestations are muscle rigidity, bradykinesia, and spasticity, predominantly affecting one side of the body. Dystonia and myoclonus are also common, as is the “alien hand” phenomenon, in which one limb moves involuntarily without conscious control. On the cognitive side, executive dysfunction, ideomotor apraxia, and language impairments such as non-fluent progressive aphasia are observed. Sensory symptoms include cortical sensory loss and tactile agnosia, making it difficult to recognize objects by touch [63].

Tau protein plays a fundamental role in the pathophysiology of CBD, as its alteration leads to hyperphosphorylation and aggregation, resulting in neuronal degeneration. In CBD, the predominant Tau isoform is 4R, which accumulates as neurofibrillary tangles and astrocytic plaques, mainly affecting the frontal and parietal cortex, basal ganglia, and substantia nigra [64,65].

Diagnosing CBD is challenging due to its overlap with other neurodegenerative disorders such as Parkinson’s disease, PSP, and frontotemporal dementia. It is based on clinical criteria and complementary studies such as magnetic resonance imaging (MRI), which may show atrophy in the parietal and frontal cortex, as well as hypointensity in the basal ganglia. Additionally, fluorodeoxyglucose positron emission tomography (PET-FDG) can detect hypometabolic patterns in the affected regions [66].

Currently, there is no curative treatment for CBD, and therapy focuses on symptom management. Levodopa and other dopaminergic agonists offer a limited response in most cases. Amantadine may be helpful in some patients with rigidity and bradykinesia. Physical and occupational therapy are essential to preserve mobility and prevent falls, while speech therapy is beneficial for improving communication in patients with aphasia [67]. In the research field, various therapeutic strategies have been explored, including anti-Tau immunotherapy with monoclonal antibodies, Tau splicing modulators, and neuroprotective drugs targeting the GSK3β kinase [68].

The prognosis of CBD is guarded, with an average survival of 6 to 8 years from symptom onset. The progression of the disease leads to a significant loss of functional independence in the early years, and falls, dysphagia, and respiratory complications are key factors in patient decline.

Research on CBD continues to progress with the aim of developing specific biomarkers to improve early diagnosis and explore therapeutic strategies targeting Tau protein. Likewise, the use of artificial intelligence to analyze clinical and neuroimaging patterns could facilitate more accurate detection of the disease in its early stages.

### 3.4. Frontotemporal Dementia with Parkinsonism Linked to Chromosome 17 (FTDP-17) and Its Relationship with Tau

Frontotemporal Dementia with Parkinsonism Linked to Chromosome 17 (FTDP-17) is a hereditary neurodegenerative disorder caused by mutations in the *MAPT* gene. This disease is part of the group of familial tauopathies and is characterized by a combination of cognitive, behavioral, and motor symptoms [69]. It is estimated that FTDP-17 accounts for between 5% and 10% of familial frontotemporal dementia cases, although its exact prevalence is unknown due to its rarity. The disease typically manifests between the ages of 40 and 60 and shows variable progression depending on the specific mutation in *MAPT* [70].

The molecular basis of FTDP-17 lies in alterations of the *MAPT* gene, located on chromosome 17q21.31. More than 50 mutations associated with this disease have been identified, which affect the structure and function of the Tau protein [71]. Some mutations alter the protein sequence, reducing its ability to bind to microtubules, while others affect messenger RNA processing, changing the proportion of Tau isoforms (3R and 4R), favoring pathological aggregation [72].

From a pathological perspective, FTDP-17 is characterized by the abnormal accumulation of hyperphosphorylated Tau in neurons and glia. The dysregulation of the 3R/4R Tau isoform balance contributes to neurodegeneration, with the formation of neurofibrillary tangles, astrocytic inclusions, and oligodendroglial bodies. These alterations mainly affect the frontal cortex, basal ganglia, hippocampus, and white matter [73].

Clinically, FTDP-17 presents with a progressive frontotemporal syndrome accompanied by parkinsonism. Cognitive symptoms include executive dysfunction, personality changes, apathy, disinhibition, and language impairments such as progressive aphasia. Episodic memory is usually less affected in the early stages of the disease. On the motor side, patients may develop parkinsonism with bradykinesia, rigidity, and postural tremor, although with limited response to levodopa. Dystonia, gait disturbances, and involuntary movements may also be present [74].

The diagnosis of FTDP-17 is based on clinical evaluation, neuroimaging studies, and genetic testing. MRI typically reveals atrophy in the frontal and temporal lobes, while FDG-PET shows hypometabolism in these regions. Additionally, DaTscan SPECT may be useful in assessing dopaminergic dysfunction in cases with parkinsonism. In cerebrospinal fluid analysis, increased total Tau levels and a reduced 3R/4R ratio have been observed. Definitive diagnosis is confirmed by sequencing the *MAPT* gene [75].

Currently, there is no curative treatment for FTDP-17, and therapy focuses on symptomatic management. Serotonin reuptake inhibitors may improve some behavioral symptoms, while levodopa and dopaminergic agonists show limited response in patients with parkinsonism. Atypical antipsychotics can be used to treat agitation and aggression, although with caution due to adverse effects.

Several therapeutic approaches are under investigation. Anti-Tau immunotherapy, based on monoclonal antibodies such as Gosuranemab and Tilavonemab, has shown potential to reduce Tau accumulation in the brain. In addition, antisense oligonucleotides (ASOs), such as BIIB080, have been shown to reduce Tau levels in cerebrospinal fluid in clinical trials. Other approaches include Tau splicing modulators and drugs targeting Tau hyperphosphorylation [76].

The prognosis of FTDP-17 varies depending on the specific mutation and symptom progression, with an average survival of between 5 and 12 years from disease onset. In advanced stages, patients require assistance with daily activities due to motor and cognitive decline.

Future research in FTDP-17 focuses on identifying specific biomarkers, developing therapies aimed at preventing Tau aggregation, and exploring genetic strategies to correct alterations in *MAPT*. Understanding the relationship between Tau and neurodegeneration in FTDP-17 could pave the way for new therapeutic interventions in both hereditary and sporadic tauopathies.

### 3.5. Pick’s Disease and Its Relationship with Tau

Pick’s Disease is a rare neurodegenerative tauopathy characterized by focal degeneration of the frontal and temporal lobes of the brain. It is considered one of the main causes of Frontotemporal Dementia (FTD) and is distinguished by the presence of Pick bodies—neuronal inclusions composed of the 3R isoform of the Tau protein. It was first described in 1892 by neurologist Arnold Pick, who identified severe brain atrophy in patients with symptoms of progressive dementia [77].

Pick’s Disease primarily affects individuals between 40 and 65 years of age, with a gradual progression leading to death within 6 to 10 years. In most cases, it is a sporadic disease, although hereditary forms exist and are associated with mutations in the *MAPT* gene, which encodes the Tau protein [78].

From a pathological perspective, Pick’s Disease differs from other tauopathies due to the accumulation of Pick bodies—spherical structures formed by aggregates of hyperphosphorylated Tau. These bodies are predominantly found in neurons of the frontal and temporal cortex, hippocampus, and caudate nucleus. Additionally, brain atrophy is more pronounced than in other dementias, leading to significant cortical volume loss and ventricular enlargement [79].

Clinically, Pick’s Disease presents with behavioral, cognitive, and language changes. The most characteristic symptoms include disinhibition, apathy, impulsivity, and hyperorality, which manifests as changes in eating behavior and compulsive consumption of certain foods, especially sweets. Repetitive behaviors and stereotyped actions are also common. Cognitively, patients experience impairments in planning, decision-making, and emotional regulation, as well as loss of empathy. In advanced stages, progression toward mutism is observed [80].

Diagnosis of Pick’s Disease is clinical, supported by neuroimaging studies. Brain MRI typically shows atrophy in the frontal and temporal lobes, while FDG-PET reveals a characteristic pattern of hypometabolism in these regions. In cerebrospinal fluid analysis, increased total Tau levels and a predominance of the 3R-Tau isoform have been observed, which helps differentiate it from other tauopathies such as Progressive Supranuclear Palsy (PSP) or Corticobasal Degeneration (CBD) [77].

Currently, there is no curative treatment for Pick’s Disease, and therapy focuses on symptomatic management. Serotonin reuptake inhibitors may help control disinhibition and impulsivity, while atypical antipsychotics are used cautiously to treat agitation and aggression. Speech and occupational therapy are essential to maintain communication and functional independence during the early stages of the disease.

In the field of research, several therapeutic strategies are under development. Anti-Tau immunotherapy with monoclonal antibodies such as Gosuranemab and Tilavonemab has shown potential in reducing Tau burden in the brain, although without significant clinical impact so far. Other approaches include the use of antisense oligonucleotides (ASOs) to modulate Tau isoform expression and therapies aimed at reducing neuroinflammation [76].

The prognosis for Pick’s Disease is poor, with rapid progression of functional decline and an average survival of 6 to 10 years from symptom onset. In advanced stages, patients require full assistance with daily activities due to dementia progression and loss of autonomy.

Future research in Pick’s Disease focuses on the development of specific biomarkers to improve early diagnosis and the identification of new therapeutic strategies targeting Tau aggregation and neuroprotection. The application of artificial intelligence in neuroimaging analysis and clinical pattern recognition could facilitate early detection and improve patient prognosis.

### 3.6. Astroglial Tauopathies and Tau

Astroglial tauopathies, also known as glial tauopathies, are a subset of primary central nervous system (CNS) tauopathies characterized by the abnormal accumulation of Tau protein in astrocytes, a type of glial cell essential for maintaining neuronal homeostasis and synaptic function [81]. Unlike classical tauopathies, in which Tau aggregates primarily within neurons, astroglial tauopathies feature Tau inclusions predominantly in astrocytes, leading to neurodegeneration indirectly through astrocytic dysfunction.

These disorders include progressive supranuclear palsy (PSP), corticobasal degeneration (CBD), glial fibrillary tangle tauopathy (GFTT), Pick’s disease, chronic traumatic encephalopathy (CTE), and tau astrogliopathy with argyrophilic grains [82]. They are unified by disruptions in Tau metabolism that favor hyperphosphorylation and aggregation within astrocytes, impairing their regulatory and supportive roles in the CNS.

At the molecular level, hyperphosphorylated Tau in astroglial tauopathies often accumulates at Ser202, Thr205, and Ser422 residues, leading to the loss of Tau’s microtubule-binding function and promoting aggregate formation [83]. These aggregates show disease-specific morphologies. For example:

In PSP, Tau-positive tufted astrocytes containing predominantly 4-repeat (4R) Tau accumulate in the midbrain, basal ganglia, and frontal cortex [84].

In CBD, argyrophilic astrocytes with 4R-Tau are primarily found in the basal ganglia and parietal cortex [85].

Pick’s disease, although primarily neuronal, also shows astrocytic inclusions with 3-repeat (3R) Tau in the frontal and temporal lobes, contributing to behavioral symptoms and frontotemporal dementia [86].

CTE presents with mixed 3R/4R Tau inclusions in perivascular astrocytes, typically in individuals exposed to repetitive head trauma, such as athletes and military veterans [86,87].

Tau astrogliopathy with argyrophilic grains involves 4R-Tau inclusions with minimal neuronal involvement and manifests as progressive motor and cognitive decline [77].

These astrocyte-specific Tau pathologies contribute to disease progression by altering glial functions such as ion buffering, neurotransmitter clearance, and maintenance of the blood–brain barrier. Consequently, astrocytic Tau pathology not only reflects the underlying tauopathy but also actively shapes the neurodegenerative landscape in each condition.

### 3.7. Frontotemporal Dementia and Tau

Frontotemporal Dementia (FTD) is a group of neurodegenerative disorders characterized by progressive atrophy of the brain’s frontal and temporal lobes. It is one of the leading causes of dementia in individuals under the age of 65, with an estimated prevalence of 10 to 20 cases per 100,000 people. Unlike Alzheimer’s disease, which primarily affects memory, FTD presents with changes in personality, behavior, and language [88]. One of its main causes is the abnormal accumulation of Tau protein in the brain, classifying it within the group of tauopathies.

FTD manifests in various clinical forms, including the behavioral variant, marked by personality and behavioral changes, and primary progressive aphasia, which affects language production and comprehension. Additionally, there are associated motor syndromes such as Corticobasal Degeneration (CBD) and Progressive Supranuclear Palsy (PSP), which share the pathological accumulation of Tau in the brain [89].

In the FTD, Tau undergoes changes that lead to its hyperphosphorylation and aggregation, resulting in neuronal dysfunction and cell death. The distribution of Tau in FTD varies according to the underlying disease. In some forms of FTD, such as Pick’s disease, there is a predominant accumulation of the 3R Tau isoform, while in PSP and CBD, the 4R isoform is mainly accumulated. These differences in Tau composition influence the clinical and pathological course of the disease [90].

FTD diagnosis is clinical and supported by auxiliary tools such as neuroimaging and biomarkers. MRI may show atrophy in the frontal and temporal lobes, while FDG-PET can detect hypometabolism in these areas. In cerebrospinal fluid, increased levels of total Tau and alterations in the ratio of 3R to 4R isoforms have been observed. In some hereditary cases, genetic analysis can reveal mutations in the *MAPT* gene, responsible for Tau production, confirming a genetic predisposition to the disease [91].

Currently, there is no curative treatment for FTD, and management focuses on symptom relief. Selective serotonin reuptake inhibitors can help improve apathy and disinhibition, while atypical antipsychotics may assist in controlling aggression and impulsivity. Physical and occupational therapy are essential in motor-affected cases to preserve mobility and prevent falls. In terms of language, speech therapy can support patients with progressive aphasia in maintaining communication for a longer period [92].

Ongoing research in FTD is focused on Tau-targeted treatments. Immunotherapy with monoclonal antibodies, such as Gosuranemab and Tilavonemab, has shown potential in reducing Tau accumulation in the brain, though clinical results are still uncertain. Other approaches include antisense oligonucleotides (ASOs), which aim to modulate Tau expression and correct the imbalance between the 3R and 4R isoforms. Additionally, neuroprotective therapies are being explored to inhibit Tau hyperphosphorylation and reduce neuroinflammation [76].

The prognosis of FTD varies depending on the clinical form and progression rate, with an average survival of 6 to 10 years from symptom onset. In advanced stages, patients require assistance with daily activities due to the progression of cognitive and motor decline.

Research in FTD continues to advance with the goal of improving early detection and developing effective therapeutic strategies. The identification of specific biomarkers and the use of artificial intelligence in neuroimaging analysis may facilitate more accurate diagnosis and allow for intervention during the early stages of the disease, offering a better quality of life for patients and their families.

### 3.8. Tau Involvement in Lysosomal Storage Diseases (LSDs)

Although Tau is classically associated with primary tauopathies like Alzheimer’s disease, growing evidence suggests its involvement in secondary tauopathies, including certain lysosomal storage diseases (LSDs). In neuropathic forms of Gaucher disease—caused by mutations in the GBA1 gene leading to glucocerebrosidase deficiency—the accumulation of undegraded substrates in lysosomes results in widespread cellular dysfunction, including neuroinflammation and neurodegeneration. Recent studies have shown that these processes can promote Tau hyperphosphorylation and aggregation, linking Gaucher disease to Tau-related pathology [93,94].

In other LSDs, such as Niemann-Pick type C and mucopolysaccharidoses, similar patterns of Tau dysregulation have been observed. Lysosomal dysfunction impairs protein clearance and can exacerbate Tau accumulation through defective autophagy and increased oxidative stress [95] Table 2.

### 3.9. Tau-Targeted Therapeutic Strategies

Tau-targeted therapies represent a promising area of investigation for the treatment of tauopathies, given Tau protein’s central role in neurodegenerative processes. These strategies aim to reduce Tau accumulation, prevent its pathological modifications, or restore its physiological function. Below are the principal therapeutic approaches under investigation:Immunotherapy: This approach uses monoclonal antibodies designed to recognize and eliminate pathological Tau. Examples include Gosuranemab and Tilavonemab, which have shown potential in reducing Tau levels in cerebrospinal fluid (CSF), although their clinical efficacy remains limited [60,76].Antisense Oligonucleotides (ASOs): ASOs are short, synthetic nucleic acid sequences that bind to Tau mRNA to reduce its translation. BIIB080 is a notable example that has shown effectiveness in lowering Tau protein levels in both preclinical and early clinical trials [76].Tau Aggregation Inhibitors: These small molecules prevent the formation of neurofibrillary tangles by inhibiting Tau aggregation. LMTX (a derivative of methylene blue) has been evaluated for this purpose, although results have been mixed in terms of clinical benefit [96].Kinase Inhibitors: Targeting kinases such as GSK-3β, which phosphorylate Tau, is another strategy. GSK-3β inhibitors aim to reduce Tau hyperphosphorylation and subsequent aggregation, but toxicity and specificity remain significant challenges [61].Splicing Modulators: These therapies aim to correct the imbalance between 3R and 4R Tau isoforms by modulating alternative splicing of the *MAPT* gene. Such modulation may be particularly relevant in disorders like PSP, CBD, and Pick’s disease [97].Neuroinflammation Modulators: Since inflammation contributes to Tau pathology, drugs that reduce microglial activation or cytokine release may have protective effects. Anti-inflammatory approaches may be used in combination with other Tau-targeted therapies [98].Chaperone-Mediated Therapy: Molecular chaperones that assist in protein folding may help prevent Tau misfolding and aggregation. Enhancing the function of these chaperones is being explored as a potential therapeutic strategy [99].

The development of these therapies is accompanied by advances in biomarker technologies, such as Tau-PET imaging and fluid biomarkers (e.g., pTau and total Tau in CSF), which help evaluate treatment efficacy in clinical trials. Although no Tau-targeted therapy has yet achieved regulatory approval, continued research is essential for translating these approaches into effective clinical interventions across the spectrum of tauopathies.

## 4. Conclusions

Tau protein, essential for the stability of the neuronal cytoskeleton, is a key element in both the physiology and pathology of the nervous system. Its expression, regulation, transport, and specific localization within neurons are finely orchestrated to maintain polarity, axonal transport, and synaptic functionality. Under physiological conditions, Tau is locally synthesized in the axon due to the specific localization of its mRNA, a process mediated by untranslated regions and multiple regulatory proteins such as HuD, FMRP, and kinesins, ensuring a precise response to cellular demands.

However, Tau also plays a central role in various neurodegenerative diseases known as tauopathies. Its dysregulation, mainly through hyperphosphorylation and abnormal aggregation, directly contributes to synaptic dysfunction, neuronal loss, and brain atrophy. In Alzheimer’s disease, the accumulation of hyperphosphorylated Tau in specific regions such as the temporal and frontal lobes is closely associated with cognitive decline, behavioral changes, and clinical progression. The asymmetric distribution of Tau in the brain, observable through techniques like Tau-PET, provides valuable clues for diagnosis and disease monitoring.

Likewise, in primary tauopathies such as Progressive Supranuclear Palsy (PSP), Corticobasal Degeneration (CBD), and Frontotemporal Dementia with Parkinsonism (FTDP-17), the involvement of different Tau isoforms—particularly 4R—has been identified, with pathological accumulations in specific brain regions and in glial cells, resulting in particular symptom profiles including movement disorders, cognitive impairment, and language deficits. Pick’s disease, in contrast, is characterized by the predominance of the 3R isoform and is marked by severe frontal and temporal involvement with profound behavioral manifestations.

The role of astrocytes and Tau-associated astrogliopathies is also a significant finding. The accumulation of Tau in glial cells—traditionally less studied than neurons—reveals that neurodegeneration is not solely a neuronal phenomenon but involves dysfunction in the glioneuronal network, which may be key to developing new therapies.

Currently, the search for curative treatments is ongoing. While symptomatic therapies exist, the most promising approaches include anti-Tau immunotherapy, splicing modulators, phosphorylation inhibitors, and antisense oligonucleotides, among others. However, their results have yet to demonstrate significant clinical impact, highlighting the need to deepen our understanding of Tau biology and its multifactorial role in neurodegeneration.

## 5. Future Perspectives

Understanding the multifaceted roles of Tau protein in both neuronal and glial compartments has significantly advanced over the past decades, yet many aspects of Tau biology and tauopathy pathogenesis remain unresolved. Future research will likely focus on deciphering the specific molecular mechanisms that differentiate physiological from pathological Tau, particularly in terms of isoform expression, post-translational modifications, and cell-type-specific vulnerability.

A major avenue for advancement lies in the development of reliable and accessible biomarkers for early diagnosis and disease progression monitoring. Tau-PET imaging and cerebrospinal fluid assays have shown promise, but their translation into widely available diagnostic tools remains limited by cost, technical complexity, and variability across tauopathies.

Therapeutic strategies are also expected to become more targeted and individualized. Novel approaches such as antisense oligonucleotides (ASOs), immunotherapies, splicing modulators, and kinase inhibitors are already under clinical evaluation, with the potential to selectively modulate Tau expression, prevent its aggregation, or restore normal microtubule dynamics. Combining these molecular therapies with personalized medicine and artificial intelligence-based diagnostic tools could enable more precise and effective treatment protocols.

In parallel, greater attention should be given to glial contributions to tauopathy pathology, particularly the role of astrocytic and microglial responses in disease onset and progression. Targeting glial Tau accumulation and the inflammatory microenvironment may open new therapeutic windows beyond neuronal-focused strategies.

Finally, the integration of multi-omics data, including genomics, transcriptomics, proteomics, and metabolomics, combined with patient-derived models such as induced pluripotent stem cells (iPSCs) and organoids, will deepen our understanding of disease mechanisms and enable high-throughput drug screening platforms.

The coming years will be critical for transitioning from descriptive neuropathology to mechanism-based, disease-modifying interventions that can halt or reverse Tau-mediated neurodegeneration.

## Figures and Tables

**Table 1 neurolint-17-00075-t001:** Characteristics of Tau Isoforms in the Nervous System.

Tau Isoform	Exon Composition	Repeat Type	Molecular Weight	Main Localization	Reference
0N3R	Lacks exons 2, 3, and 10	3R	~37 kDa	Fetal brain, adult cortex	[5]
1N3R	Includes exon 2 only	3R	~39 kDa	Fetal and adult brain (low levels)	[5]
2N3R	Includes exons 2 and 3	3R	~45 kDa	Adult cortex	[5]
0N4R	Includes exon 10 only	4R	~62 kDa	Adult brain (basal ganglia, brainstem)	[5]
1N4R	Includes exons 2 and 10	4R	~64 kDa	Adult cortex and subcortical regions	[5]
2N4R	Includes exons 2, 3, and 10	4R	~70 kDa	Adult brain, especially hippocampus	[5]
Big Tau	Includes exon 4A	4R	~110 kDa	Peripheral nervous system	[7]

**Table 2 neurolint-17-00075-t002:** Overview of Tauopathies and Their Clinical Characteristics.

Tauopathy	Key Characteristics	Dominant Tau Isoform	Main Affected Regions	Reference(s)
Alzheimer’s Disease (AD)	Most common tauopathy associated with memory loss, cognitive decline	Mixed (3R/4R)	Temporal and frontal lobes	[35,40]
Progressive Supranuclear Palsy (PSP)	Vertical gaze palsy, postural instability, axial rigidity	4R	Midbrain, basal ganglia, frontal cortex	[55,56]
Corticobasal Degeneration (CBD)	Asymmetric parkinsonism, ‘alien hand’ syndrome	4R	Frontal/parietal cortex, basal ganglia	[64,65]
Frontotemporal Dementia with Parkinsonism (FTDP-17)	Genetic, behavioral and motor symptoms	3R/4R (variable)	Frontal cortex, basal ganglia	[71,74]
Pick’s Disease	Behavioral changes, language impairment, Pick bodies	3R	Frontal and temporal lobes	[77,78]
Astroglial Tauopathies	Tau in astrocytes, linked with neuroinflammation	Mainly 4R	Frontal cortex, basal ganglia, cortex	[81,82]
Lysosomal Storage Disorders (e.g., Gaucher)	Secondary tauopathy, Tau aggregation with lysosomal dysfunction	Variable	Temporal and cortical regions	[93,94]

## Data Availability

Not applicable.

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
