# Peer review of "The Role of Tau in Neuronal Function and Neurodegeneration"

_2035-8377, 2025, doi:10.3390/neurolint17050075_

Round 1
Reviewer 1 Report
Comments and Suggestions for Authors
The subject matter of the manuscript is appropriate and highly relevant. However, the text currently lacks coherence, with several sections appearing disjointed and highly repetitive. The manuscript would benefit from extensive and careful revision to enhance clarity, logical flow, and overall readability.
Specific Comments and Suggestions:
-To reduce redundancy, consider adding a brief sentence after line 158 clarifying that progressive supranuclear palsy (PSP), corticobasal degeneration (CBD), and similar disorders are classified as tauopathies. This would eliminate the need to restate this information in each disease-specific subsection.
-Section 3.1, which addresses Alzheimer’s disease, should be revised or potentially rewritten. At present, the paragraphs lack a clear logical flow, which makes it difficult to fully appreciate the role and significance of tau in the context of AD. Given that Alzheimer’s disease is the most prevalent tauopathy, improving the structure and coherence of this section would strengthen the manuscript significantly.
-The section on astroglial tauopathy should also be rewritten
-The authors are encouraged to include a dedicated section on therapies targeting tau, as these strategies are generally applicable across all tauopathies. Centralizing this information, rather than repeating it at the end of each disease-specific subsection, would streamline the manuscript and reduce repetition.
-Several references should be reviewed for accuracy and relevance. A few examples but there are others:
- In line 122, reference 26 should be replaced, as it does not provide evidence for the modulation of Tau mRNA localization and translation by BDNF or NGF.
- In line 175, reference 37 does not adequately support the claim that “Tau protein is closely associated with Alzheimer’s disease-related pathologies.” A more appropriate citation should be selected, or additional context provided to justify the statement.
-Although clarified later in the section, the sentence “…present in the 3’ UTR regions of certain mRNAs [30]” would benefit from the addition of “, including tau.” This would help establish the relevance of HuD earlier in the paragraph.
-Lines 509–514 and 561–563 contain repeated explanations of tau protein and its characteristics, which are redundant and should be removed or consolidated.
Minor Comment:
Line 346: "CBS" should be corrected to "CBD."
Author Response
Dear Reviewer 1.
We thank Reviewer 1 for his time in reviewing the manuscript and for the important suggestions and comments he has given us to improve the article.
The subject matter of the manuscript is appropriate and highly relevant. However, the text currently lacks coherence, with several sections appearing disjointed and highly repetitive. The manuscript would benefit from extensive and careful revision to enhance clarity, logical flow, and overall readability.
Specific Comments and Suggestions:
-To reduce redundancy, consider adding a brief sentence after line 158 clarifying that progressive supranuclear palsy (PSP), corticobasal degeneration (CBD), and similar disorders are classified as tauopathies. This would eliminate the need to restate this information in each disease-specific subsection.
Thank you very much for the suggestion, the sentence has been added.
-Section 3.1, which addresses Alzheimer’s disease, should be revised or potentially rewritten. At present, the paragraphs lack a clear logical flow, which makes it difficult to fully appreciate the role and significance of tau in the context of AD. Given that Alzheimer’s disease is the most prevalent tauopathy, improving the structure and coherence of this section would strengthen the manuscript significantly.
Thank you very much for the suggestion, this section was rewritten.
-The section on astroglial tauopathy should also be rewritten
Thank you very much for the suggestion, this section was rewritten.
-The authors are encouraged to include a dedicated section on therapies targeting tau, as these strategies are generally applicable across all tauopathies. Centralizing this information, rather than repeating it at the end of each disease-specific subsection, would streamline the manuscript and reduce repetition.
Thank you very much for the suggestion, the section was added in the manuscript.
-Several references should be reviewed for accuracy and relevance. A few examples but there are others:
- In line 122, reference 26 should be replaced, as it does not provide evidence for the modulation of Tau mRNA localization and translation by BDNF or NGF.
- In line 175, reference 37 does not adequately support the claim that “Tau protein is closely associated with Alzheimer’s disease-related pathologies.” A more appropriate citation should be selected, or additional context provided to justify the statement.
Thank you for your comment, the references have been updated.
-Although clarified later in the section, the sentence “…present in the 3’ UTR regions of certain mRNAs [30]” would benefit from the addition of “, including tau.” This would help establish the relevance of HuD earlier in the paragraph.
Thanks for the suggestion, the change in the sentence has been made.
-Lines 509–514 and 561–563 contain repeated explanations of tau protein and its characteristics, which are redundant and should be removed or consolidated.
Thanks for the suggestion, it was removed.
Minor Comment:
Line 346: "CBS" should be corrected to "CBD."
Thank you very much, the change has been made.

Reviewer 2 Report
Comments and Suggestions for Authors
The review “The Role of Tau in Neuronal Function and Neurodegeneration:From Axonal Transport to Tauopathies” by Gonzalo Emiliano Aranda-Abreu describes role of Tau protein in normal neurophisiology and neurodegenerative conditions such as Alzheimer’s disease, Progressive Supranuclear Palsy, Corticobasal Degeneration, Pick’s Disease, astroglial tauopathies, Frontotemporal Dementia and others. The review is comprehensive and in my opinion can be published in its current state, with minor changes. Please find below my comments and suggestions
Firstly, all gene names should be italicized.
Line 24 quote:“effective disease-modifying therapies remain elusive”. - If I were author, I would have added words “for tauopaties” in this sentence.
Line 65. quote: “number of repeats” - what repeats? Please explain in more detail
Line 76. quote: “the MAPT gene (microtubule-associated protein tau), which encodes the Tau protein isoforms” - no need to explain it here, it should be explained in line 59.
In Table 1, could you please provide references?
Line 86. quote: “CREB, a cAMP response element-binding protein, has also been identified” - I would have said “ has also been identified among regulators of MAPT gene expression”.
Line 119. quote: “miRNA Binding Motifs, such as miR-132” - miR-132 is not a miRNA binding motif, it’s a miRNA. There are miRNA binding motifs, such as miR-132 binding motifs. Perhaps the sentence should be rewritten.
Line 207. Please introduce/briefly describe Soluble Triggering receptor expressed on myeloid cells 2 (sTREM2), mentioned in this sentence.
Line 391.quote: “MAPT gene (Microtubule-Associated Protein Tau), which encodes the Tau protein” - this info is redundant, see. Lines 76, 59.
Line 512. quote: “In astroglial tauopathies, hyperphosphorylated Tau is found at residues such as Ser202, Thr205, and Ser422” - I would have said that “In astroglial tauopathies Tau is hyperphosphorylated at residues Ser202, Thr205, and Ser422”
Speaking about phosphorylated Tau (p-Tau), what other posttranslational modifications of Tau are known and what are their functional roles? Perhaps you can briefly discuss role of Tau’s Acetylation, Methylation, Glycosylation and Glycation, Proteolysis, Oxydation and Nitration, etc.
The review is entitled “ <...> From Axonal Transport to Tauopathies”, however tha main focus of the review is on Tauopathies. Could you please add more info on the role of Tau in Axonal transport (AT), in both normal conditions and neurodegereration, perhaps in a form of the schematic illustration summarizing such info?
You might also want to discuss in more details how impalance of Tau Isoforms affects AT of the Amyloid Precursor Protein in neurons.
What about the role of Tau in neuropathic forms of some hereditary diseases from the “umbrella” of Lysosomal storage diseases (LSD) such as Gaucher Disease? What about involvement of Tau in neurodegeneration observed in other LSDs?
Author Response
Dear Reviewer 2
We thank Reviewer 2 for his time in reviewing the manuscript and for the important suggestions and comments he has given us to improve the article.
The review “The Role of Tau in Neuronal Function and Neurodegeneration:From Axonal Transport to Tauopathies” by Gonzalo Emiliano Aranda-Abreu describes role of Tau protein in normal neurophisiology and neurodegenerative conditions such as Alzheimer’s disease, Progressive Supranuclear Palsy, Corticobasal Degeneration, Pick’s Disease, astroglial tauopathies, Frontotemporal Dementia and others. The review is comprehensive and in my opinion can be published in its current state, with minor changes. Please find below my comments and suggestions
Firstly, all gene names should be italicized.
Thank you very much for the suggestion, the changes were made.
Line 24 quote:“effective disease-modifying therapies remain elusive”. - If I were author, I would have added words “for tauopaties” in this sentence.
Thanks for the suggestion, the sentence was added.
Line 65. quote: “number of repeats” - what repeats? Please explain in more detail
Thanks for the suggestion, the sentence has been rewritten.
Line 76. quote: “the MAPT gene (microtubule-associated protein tau), which encodes the Tau protein isoforms” - no need to explain it here, it should be explained in line 59.
Thanks for the suggestion, the change was made.
In Table 1, could you please provide references?
Thank you, the references have been added.
Line 86. quote: “CREB, a cAMP response element-binding protein, has also been identified” - I would have said “ has also been identified among regulators of MAPT gene expression”.
Thank you, the change has been made.
Line 119. quote: “miRNA Binding Motifs, such as miR-132” - miR-132 is not a miRNA binding motif, it’s a miRNA. There are miRNA binding motifs, such as miR-132 binding motifs. Perhaps the sentence should be rewritten.
Thanks for the suggestion, the sentence was rewritten.
Line 207. Please introduce/briefly describe Soluble Triggering receptor expressed on myeloid cells 2 (sTREM2), mentioned in this sentence.
Thank you, sTREM2 was described.
Line 391.quote: “MAPT gene (Microtubule-Associated Protein Tau), which encodes the Tau protein” - this info is redundant, see. Lines 76, 59.
Thank you.
Line 512. quote: “In astroglial tauopathies, hyperphosphorylated Tau is found at residues such as Ser202, Thr205, and Ser422” - I would have said that “In astroglial tauopathies Tau is hyperphosphorylated at residues Ser202, Thr205, and Ser422”
Thanks for the suggestion, the section was rewritten.
Speaking about phosphorylated Tau (p-Tau), what other posttranslational modifications of Tau are known and what are their functional roles? Perhaps you can briefly discuss role of Tau’s Acetylation, Methylation, Glycosylation and Glycation, Proteolysis, Oxydation and Nitration, etc.
Excellent suggestion. In this review, we focus particularly on hyperphosphorylation. It would be very important in another review to write about the different posttranslational modifications. In fact we are working on it, with molecular modeling.
The review is entitled “ <...> From Axonal Transport to Tauopathies”, however tha main focus of the review is on Tauopathies. Could you please add more info on the role of Tau in Axonal transport (AT), in both normal conditions and neurodegereration, perhaps in a form of the schematic illustration summarizing such info?
Thanks for the suggestion. The topic of axonal transport under normal conditions and neurodegeneration is very extensive, which would give scope for a new manuscript. In the section of the article we briefly describe axonal transport and the molecules involved in the normal way. Indeed, the article is more focused on thaupathies.
Based on this, we will change the title, deleting. From Axonal Transport to Tauopathies
You might also want to discuss in more details how impalance of Tau Isoforms affects AT of the Amyloid Precursor Protein in neurons.
What about the role of Tau in neuropathic forms of some hereditary diseases from the “umbrella” of Lysosomal storage diseases (LSD) such as Gaucher Disease? What about involvement of Tau in neurodegeneration observed in other LSDs?
Thanks for the suggestion, we will add a section on LSD.

Round 2
Reviewer 1 Report
Comments and Suggestions for Authors
Section 3.1 remains unfocused and lacks coherence, with a mixture of information that does not clearly support the central topic. It is unclear why frontotemporal lobar degeneration (FTLD) is discussed in Lines 222–227, as this appears out of place in a section intended to focus on Alzheimer’s disease. Additionally, the relevance of the content in Lines 211–215 and 237–242 to AD is unclear. The authors should revise this section to ensure it presents a clear, structured discussion specifically related to Alzheimer’s disease, avoiding tangential information unless its relevance is explicitly explained.
Section 2. Lines 110–123: The section discussing cis-regulatory elements in the UTRs of tau is misleading, as the statements made are not adequately supported by the cited references. Specifically, reference 26 does not substantiate the claim that "response elements to these stimuli located in the 3’ UTR allow tau mRNA localization and translation to be modulated by extracellular factors, such as BDNF or NGF, through intracellular signaling cascades." Additionally, reference 27 is overly general and does not provide information specific to tau. This section should be revised to more accurately reflect what is known about tau-specific regulatory mechanisms, with appropriately tailored references or more cautious language.
Section 3.6 on Astroglial Tauopathies: This section still requires extensive editing for clarity and focus. It is often unclear whether the discussion pertains to tauopathies in general or specifically to the role of glial cells—for example, in Lines 458–463. The sentence in Lines 448–450 is unclear and potentially misleading, and the cited reference does not adequately support the statement made. Additionally, the section contains a significant amount of disjointed information. For instance, Lines 472–476 do not appear to relate to astroglia and seem out of place in this context. A more structured and topic-focused revision is necessary to improve readability and relevance.
Author Response
Dear Reviewer 1,
Thank you for your comments and very helpful suggestions to the manuscript. The changes done are highlighted in green color.
Section 3.1 remains unfocused and lacks coherence, with a mixture of information that does not clearly support the central topic. It is unclear why frontotemporal lobar degeneration (FTLD) is discussed in Lines 222–227, as this appears out of place in a section intended to focus on Alzheimer’s disease. Additionally, the relevance of the content in Lines 211–215 and 237–242 to AD is unclear. The authors should revise this section to ensure it presents a clear, structured discussion specifically related to Alzheimer’s disease, avoiding tangential information unless its relevance is explicitly explained.
- Thank you very much, I totally agree with what you mention. The FTLD part was removed and the section was rewritten in a better way.
Section 2. Lines 110–123: The section discussing cis-regulatory elements in the UTRs of tau is misleading, as the statements made are not adequately supported by the cited references. Specifically, reference 26 does not substantiate the claim that "response elements to these stimuli located in the 3’ UTR allow tau mRNA localization and translation to be modulated by extracellular factors, such as BDNF or NGF, through intracellular signaling cascades." Additionally, reference 27 is overly general and does not provide information specific to tau. This section should be revised to more accurately reflect what is known about tau-specific regulatory mechanisms, with appropriately tailored references or more cautious language.
- Thank you very much, added about the cis-regulatory elements in the UTR, it is an uracil-rich region called fragment H, added the reference that supports it. Also added the reference supporting that NGF could modulate tau localization and translation. In this case, PC12 cells treated with NGF were used.
Section 3.6 on Astroglial Tauopathies: This section still requires extensive editing for clarity and focus. It is often unclear whether the discussion pertains to tauopathies in general or specifically to the role of glial cells—for example, in Lines 458–463. The sentence in Lines 448–450 is unclear and potentially misleading, and the cited reference does not adequately support the statement made. Additionally, the section contains a significant amount of disjointed information. For instance, Lines 472–476 do not appear to relate to astroglia and seem out of place in this context. A more structured and topic-focused revision is necessary to improve readability and relevance.
R. Thank you very much, we rewrote this section and removed the sentences that did not fit the section. The main idea of the section is how Tau is able to affect glia, as it happens in neurons.
Round 3
Reviewer 1 Report
Comments and Suggestions for Authors
Thank you for addressing my comments.